# Study on the Coupled Heat Transfer Model Based on Groundwater Advection and Axial Heat Conduction for the Double U-Tube Vertical Borehole Heat Exchanger

**Linlin Zhang [1], Zhonghua Shi [2] and Tianhao Yuan [1,*]** 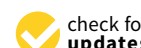

[1] School of Environmental and Municipal Engineering, North China University of Water Resources and Electric Power, Zhengzhou 450046, China; zhanglinlin@ncwu.edu.cn

[2] School of Mechanical and Electrical Engineering, Zhengzhou Vocational University of Information and Technology, Zhengzhou 450046, China; shizhonghua72@sohu.com

[*] Correspondence: ytianhao@ncwu.edu.cn

**Abstract:** In this paper, a dynamic heat transfer model for the vertical double U-tube borehole heat exchanger (BHE) was developed to comprehensively address the coupled heat transfer between the in-tube fluid and the soil with groundwater advection. A new concept of the heat transfer effectiveness was also proposed to evaluate the BHE heat exchange performance together with the index of the heat transfer rate. The moving finite line heat source model was selected for heat transfer outside the borehole and the steady-state model for inside the borehole. The data obtained in an on-site thermal response test were used to validate the physical model of the BHE. Then, the effects of soil type, groundwater advection velocity, inlet water flow rate, and temperature on the outlet water temperature of BHE were explored. Results show that ignoring the effects of groundwater advection in sand gravel may lead to deviation in the heat transfer rate of up to 38.9% of the ground loop design. The groundwater advection fosters the heat transfer of BHE. An increase in advection velocity may also help to shorten the time which takes the surrounding soil to reach a stable temperature. The mass flow rate of the inlet water to the BHE should be more than 0.5 kg·s$^{-1}$ but should not exceed a certain upper limit under the practical engineering applications with common scale BHE. The efficiency of the heat transfer of the double U-tube BHE was determined jointly by factors such as the soil's physical properties and the groundwater advection velocity.

**Keywords:** dynamic heat transfer model; borehole heat exchanger; groundwater advection; physical properties of the soil

## 1. Introduction

Due to the advantages of high efficiency, energy-saving, and environmental friendliness, the ground source heat pump (GSHP) system has been widely used in the world and become a hotspot in clean energy study in recent years [1–6]. The GSHP system has a variety of applications such as obtaining the hot water and heating/cooling the commercial and domestic space [7]. These systems have been applied in military complexes, sports centers, institutional buildings, offices, hops, hotels, and schools [8]. The structure of a GSHP system is typically divided into two main elements: a borehole heat exchanger (BHE) and a heat pump system [9]. The schematic diagram of the GSHP system used for heating/cooling is shown in Figure 1. The heat pump is a mechanical device that transfers heat energy from the ground to a building, and vice versa. Vertical BHE with single or double U-tubes are the most widely adopted forms of BHE in practice because they occupy less space and exhibit good performance [10,11]. The

depth of the vertical borehole is from 100 m to 150 m for the GSHP system [7,12]. The proper design of a BHE in the GSHP system is vital because the BHE undertakes the important processes of heat rejection to or extraction from the ground. Therefore, the heat transfer mechanism and mathematical models of BHE have attracted the interest of numerous researchers in the field of GSHP systems. Numerous numerical and analytical studies have been carried out to reveal the heat transfer processes of BHE. Classical analytical solutions include a line and cylindrical heat source [13–17]. Numerical models are based on finite volume, finite difference, or finite element methods [18–20]. The numerical model can offer discrete solutions but it takes a relatively long computational time. The analytical model is preferred in most practical applications because of its superior computational speed and flexibility for parameterized design. Infinite and finite line source analytical models can be applied in the analysis of ground temperature response, however, the heat transfer process inside the borehole is ignored.

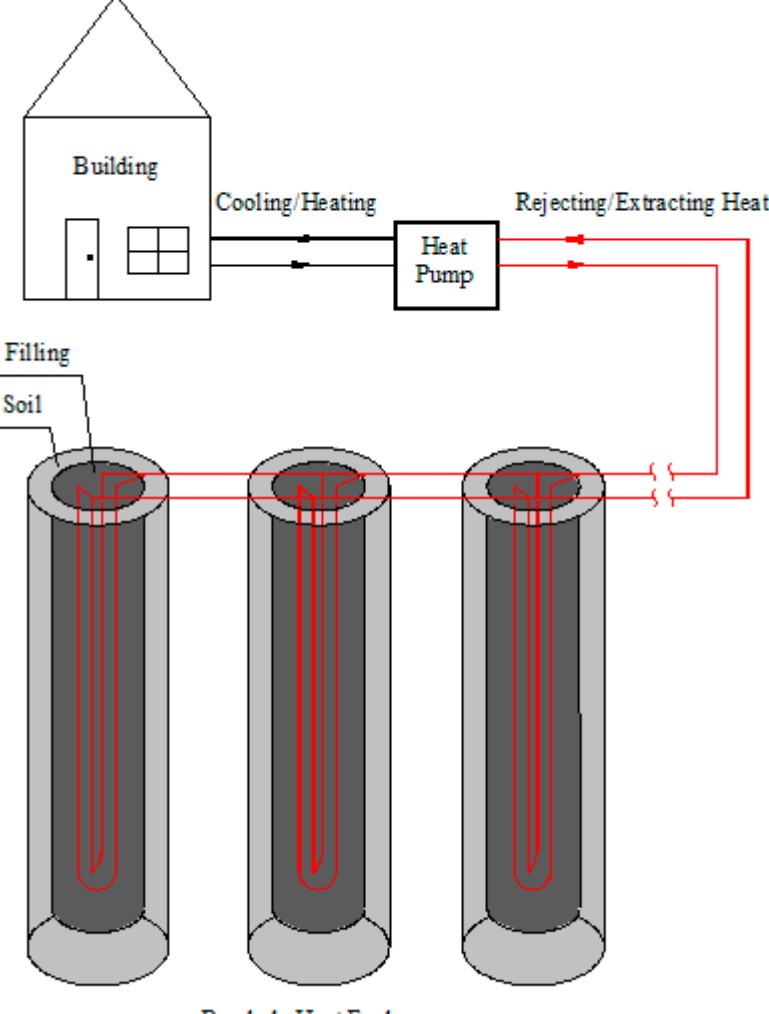

**Figure 1.** A schematic diagram of the ground source heat pump (GSHP) system used for heating/cooling.

The heat transfer performance of BHE is determined by the parameters of the media inside the tubes and the thermal physical properties of the surrounding soils. Assessing the heat transfer performance of the BHE accurately is also crucial to the design optimization of the entire system [21]. Commercial tools have been developed for BHE design, including GHLEPRO, EED, RETScreen, etc. [22], which are all based on the hypothesis of pure heat conduction in soils [13]. However, pure heat conduction in soils may lead to a significant deviation in the entire design borehole depth from the actual one required because of the neglect of groundwater advection in soil [23,24]. As a result, the initial

cost of the borehole drilling will increase in the case of groundwater advection in the soil, which may have positive or negative effects on the operational performance of a GSHP system [8].

Therefore, we decided to establish a coupled analytical model for BHE after balancing the pros and cons between numerical and analytical models to study the heat transfer process between the BHE and ground soil considering its groundwater advection.

## 2. State of the Art

Groundwater advection can be commonly found in many geological environments and significantly affects heat transfer performance around boreholes [25–27]. Existing literature has mostly analyzed the thermal performance of BHE through numerical and analytical methods that take groundwater advection into account and these studies have proved that it has a positive effect on heat transfer enhancement as the temperature rises and reaches a steadier condition [9]. Gehlin et al. [28] explored the heat transfer process between buried tubes and a fractured rock surrounding, wherein groundwater advection played a role, in continuum and discrete approaches. Even very low groundwater flow rates have been found to cause a significant enhancement in heat transfer ability. Fan et al. [29] set up a dynamic mathematical model for a vertical dual-function BHE that considered the effects of groundwater advection in the soil and concluded that the presence of groundwater had a significant effect on the heat transfer between the BHE and the soil, which could lead to an improvement in the GSHP system performance. Zanchini et al. [30] explored the effects of groundwater advection on the long-term performance of large BHE fields with unbalanced loads in the summer and the winter using the finite-element simulation method and found that even very low advection velocity could significantly promote the long-term heat transfer performance of buried tubes. Zhang et al. [24] proposed an analytical solution to soil temperature response to heat rejection from BHE with groundwater advection and suggested that groundwater advection and borehole layout should be important considerations in the design of the BHE configurations. You et al. [31] evaluated the heat exchange performance of a pile heat exchanger in a groundwater-rich area and found that the growth rate of average inlet and outlet temperatures decreased as groundwater flow velocity increased, and that the comprehensive thermal conductivity coefficient also increased. Li et al. [32] built a 3D numerical model that considered ground stratification and groundwater advection and determined that groundwater had positive effects on the tubes in the upper stream rather than those in the lower stream. In summary, these studies indicate that groundwater advection, even when not obvious in the soil, may cause a significant difference in comparison to the null velocity case.

Thus, considering the effects of groundwater advection is important in the improvement of BHE design accuracy and in the optimization of the system operation performance at the design stage of a GSHP system. Although a considerable number of studies have been conducted, the ground temperature response caused by the heat dissipation of buried tubes in the soil with groundwater advection and its effects on the outlet water temperature from BHE have seldom been explored qualitatively. Therefore, the current in-depth studies are not sufficient to reveal the effects of groundwater advection on the heat transfer performance of BHE in different constructive forms in soil under different hydrogeological conditions. With the existing BHE analytical models that use the effects of groundwater advection, the coupled problem of heat transfer from the in-tube fluid to the soil with groundwater advection cannot be solved comprehensively.

The rest of this study is organized as follows. Section 3 describes the methodology and model validation. A set of mathematical models of BHE with double U-tube was developed based on the Green's function of the moving finite line source and the steady-state model inside the borehole, taking the axial heat conduction and the effects of groundwater advection into consideration. In Section 4, the heat transfer performance of the BHE under the effects of groundwater advection was assessed in terms of the heat transfer efficiency of the BHE and the heat transfer rate to provide accurate theoretical guidance for future BHE designers. Section 5 is the conclusions.

## 3. Methodology

The thermal behavior of the BHE is complicated to model because of the various heat transfer mechanisms inside and outside the BHE. Heat transfer outside the borehole can be mainly described with heat conduction and advection between BHE and ground soil when the groundwater advection in soil is considered. Heat transfer inside the borehole can be decomposed into the following mechanisms: forced convection of circulating fluid inside the U-tube, conductive heat transfer in the tube wall and the borehole filling material, and the thermally induced natural convection [33]. The main focus of this study was the heat transfer of the double U-tube vertical BHE with groundwater advection based on the coupled heat transfer model. The model coupled with the heat transfer inside and outside the borehole was established through the value of the borehole wall temperature $T_b$ by using MATLAB. The geometry model shown in Figure 2 illustrates the dimensions and boundary conditions.

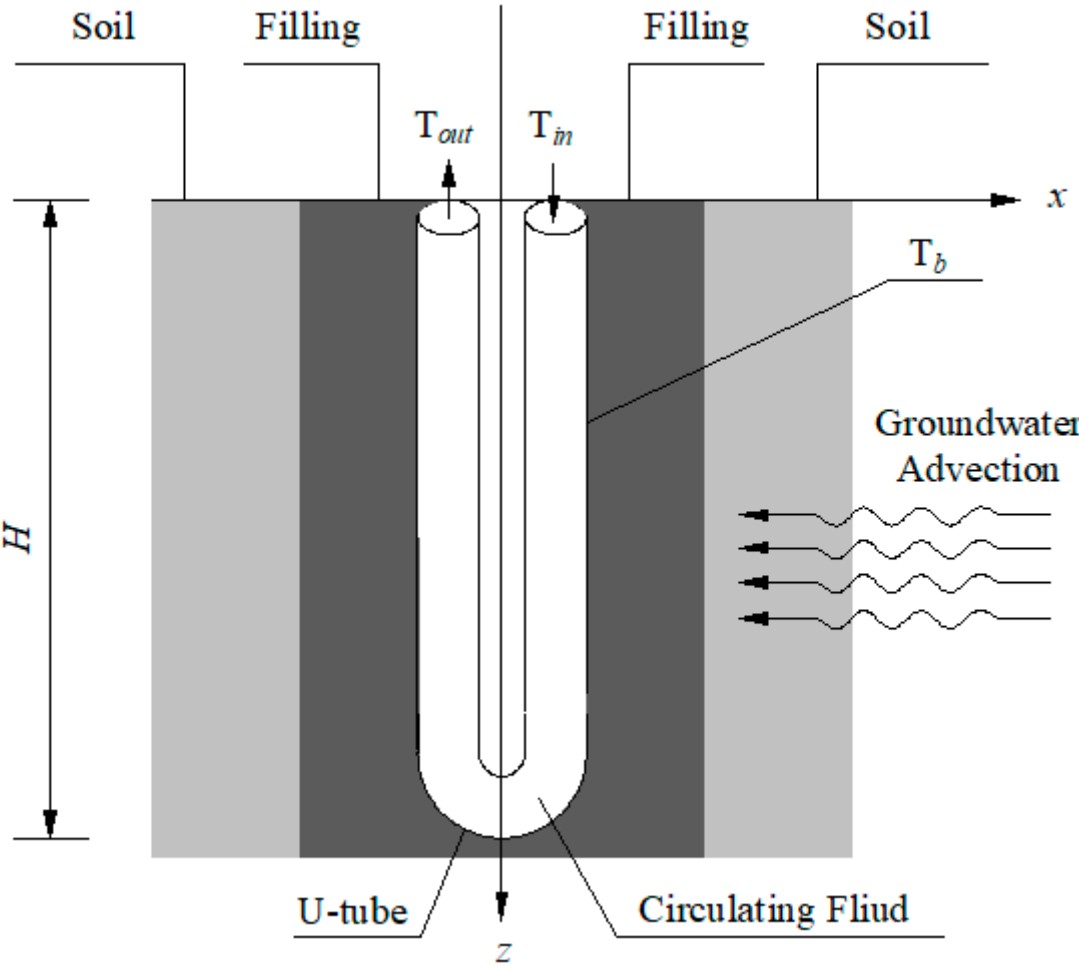

**Figure 2.** The geometric model of borehole heat exchanger (BHE).

### 3.1. Coupled Heat Transfer Model of Porous Medium inside and outside BHE

In general, heat is transferred from the water circulating in the U-tube to the tube wall by convection and then conducted through the backfilling materials inside a borehole to the surrounding soil. The line source heat transfer model was proposed by Ingersoll and Plass [34] as the design basis for the GSHP system. The heat transfer process can be treated as a line heat source that rejects or extracts energy along the vertical direction of the borehole at a constant rate only by heat conduction in a homogeneous, isotropic infinite medium [35]. This model is simple and convenient to be used in designing BHE. However, the ground surface temperature may affect the BHE heat transfer performance because the diameter of a vertical borehole is far less than its depth, despite its limitations. The physical properties

of the soil considering groundwater advection have a significant effect on this process because they will lead to a change in the outlet water temperature. In this paper, the outside BHE heat transfer process is simplified as a moving finite line source in the semi-infinite soil medium to take the effects of the ground surface temperature and the ground advection into account [13,24]. Thus, the moving finite line source model (MFLS) is established based on the following assumptions:

(1) The soil is assumed to be a homogeneous and porous medium, which is initially at thermal equilibrium and its thermal properties are independent of the temperature changes.

(2) The soil surface temperature remained constant at the initial value and its properties are independent of vertical geothermal gradient and temperature variations.

(3) Heat transfer between two legs of U-tube is negligible and a constant heat flow rate of the borehole is treated as a line source of finite length, which stretches along the $z$-axis down to a certain depth $H$ of borehole, as shown in Figure 2.

(4) All the material properties related to the BHE are temperature-independent and remain constant.

Heat transport in the porous medium soil with groundwater advection is mainly accomplished by conduction through the fluid and solid phase and advection through the flowing water. Groundwater advection was deemed as a one-dimensional uniform flow and the equivalent advection velocity $U$ in the soil was taken as the value of $U = u\rho_w c_w / \rho c$, where $u$ is the groundwater advection velocity which flows along $x$-direction as in Figure 2 (m·s$^{-1}$), $\rho_w c_w$ is volume specific heat capacity of water (J·m$^{-3}$·K$^{-1}$), $\rho c$ is the volume specific heat capacity of porous medium soil (J·m$^{-3}$·K$^{-1}$). The temperature $T'$ (K) in the coordinates is $(\xi, \eta, \mu)$ that moves with a fixed medium; the equation of heat conduction except the domain where the heat source locates is indicated as [36]:

$$\frac{1}{a}\frac{\partial T'}{\partial \tau} = \left(\frac{\partial^2 T'}{\partial^2 \xi} + \frac{\partial^2 T'}{\partial^2 \eta} + \frac{\partial^2 T'}{\partial^2 \mu}\right) \tag{1}$$

If the motionless coordinate is set as $(x, y, z)$, the relevant expression about two different coordinates are denoted as: $x = \xi + u\tau, y = \eta, z = \mu$, and the equation $T'(\xi, \eta, \mu, \tau)$ can be changed into $T(x, y, z, \tau)$. The heat diffusion process of this finite line source with constant heat emission rate located along the $z$-axis in the porous soil with groundwater advection can be depicted according to the partial differential equation as follows [24]:

$$\frac{1}{a}\frac{\partial T}{\partial \tau} = \left(\frac{\partial^2 T}{\partial^2 x} + \frac{\partial^2 T}{\partial^2 y} + \frac{\partial^2 T}{\partial^2 z}\right) - \frac{U}{a}\frac{\partial T}{\partial x} \tag{2}$$

Subject to the initial and boundary conditions, it is as follows:

$$\left.\begin{array}{l} \tau \geq 0, z = 0 : T = T_0 \\ \tau = 0, 0 < r < \infty : T = T_0 \\ \tau > 0, r \to \infty : T = T_0 \\ \tau > 0, r' \to 0 : -\lambda \frac{\partial T}{\partial r} 2\pi r = \frac{c_p M(T_{in} - T_{out})}{H} \end{array}\right\} \tag{3}$$

where $\tau$ is time (s); $T_0, T_{in}, T_{out}$ are initial temperature, inlet water temperature, and outlet water temperature (K); $a = \lambda/\rho c$ is the porous medium soil thermal diffusivity (m$^2$·s$^{-1}$); $M$ is mass flow rate of the water in the buried tubes (kg·s$^{-1}$); $c_p$ is specific heat of the water (J·kg$^{-1}$·K$^{-1}$); $H$ is borehole depth (m); $r' = \sqrt{x^2 + y^2}$ is the radius of borehole section (m); $r = \sqrt{x^2 + y^2 + (z - h)^2}$ is distance of any position (m); and $h$ is an increase in the depth of borehole (m). The effective thermal conductivity $\lambda$ (W·m$^{-1}$·K$^{-1}$) and the volume specific heat capacity of porous medium soil $\rho c$ are weighted averages of the saturated water and solid matrix, usually determined on basis of its porosity $\varepsilon$ as follows [37]: $\lambda = \varepsilon \lambda_w + (1 - \varepsilon)\lambda_s$ and $\rho c = \varepsilon \rho_w c_w + (1 - \varepsilon)\rho_s c_s$. $\rho_s c_s$ is volume specific heat capacity of soil matrix (J·m$^{-3}$·K$^{-1}$).

The solution to the dynamic temperature variations in the soil can be obtained by using the Green's function and a fictitious line heat source method, expressed by reference [13]:

$$\Delta T(x, y, z, \tau) = \frac{c_p M(T_{in} - T_{out})}{2\pi\lambda H} \exp\left(\frac{Ux}{2a}\right)\left[\int_0^H f(x, y, z, \tau)dh - \int_{-H}^0 f(x, y, z, \tau)dh\right] \quad (4)$$

where

$$f(x, y, z, \tau) = \frac{1}{4r}\left[\exp(-\frac{Ur}{2a}) \times erfc\left(\frac{r - U\tau}{2\sqrt{a\tau}}\right) + \exp(\frac{Ur}{2a}) \times erfc(\frac{r + U\tau}{2\sqrt{a\tau}})\right] \quad (5)$$

As $\tau \to \infty$, the temperature variation may reach a stable value expressed by

$$\Delta T(x, y, z) = \frac{c_p M(T_{in} - T_{out})}{4\pi\lambda H} \exp\left(\frac{Ux}{2a}\right) \times \left[\int_0^H \frac{1}{r} \exp\left(-\frac{Ux}{2a}\right)dh - \int_{-H}^0 \frac{1}{r} \exp\left(-\frac{Ux}{2a}\right)dh\right] \quad (6)$$

where the outlet water temperature of the buried U-tube, $T_{out}$, is determined by the coupled heat transfer processes inside and outside the borehole.

The quasi-three-dimensional model for the inside BHE heat transfer with a double U-tube is established to take into account the water temperature variation in the borehole. In order to keep the problem analytically manageable, the simplifications are explained as follows [38]: the heat capacity of the materials inside the borehole is neglected; the ground outside the borehole and backfilling material in it are homogeneous, and all the thermal properties involved are independent of temperature; the heat convection of the water inside and the heat conduction through the buried tubes and backfilling material were considered to be steady processes because the dimensions and thermal capacity of the U-tubes combined with the backfilling materials inside the borehole were relatively small compared with those of the surrounding ground; the axial heat conduction of backfilling materials in the borehole can be neglected and the borehole wall temperature along the borehole depth remains the same, but may vary with time.

The average temperature of the borehole wall ($T_b$) can be chosen as the reference temperature. Number the tubes in the borehole clockwise as shown in Figure 3. The thermal resistance between the water in each U-tube leg and the borehole wall is denoted as $R'_{11}$(m·K·W$^{-1}$), that between the two adjacent legs is denoted as $R'_{12}$(m·K·W$^{-1}$), and that between two symmetric legs is denoted as $R'_{13}$ (m·K·W$^{-1}$). The circulating water temperature in four branch tubes ($T_{f1}, T_{f2}, T_{f3}, T_{f4}$, K) satisfies the following equation [38]:

$$\begin{cases} T_{f1} - T_b = R'_{11}q_1 + R'_{12}q_2 + R'_{13}q_3 + R'_{12}q_4 \\ T_{f2} - T_b = R'_{12}q_1 + R'_{11}q_2 + R'_{12}q_3 + R'_{13}q_4 \\ T_{f3} - T_b = R'_{13}q_1 + R'_{12}q_2 + R'_{11}q_3 + R'_{12}q_4 \\ T_{f4} - T_b = R'_{12}q_1 + R'_{13}q_2 + R'_{12}q_3 + R'_{11}q_4 \end{cases} \quad (7)$$

where $q_1, q_2, q_3, q_4$ are the heat flux rate of each branch tube, respectively (W·m$^{-1}$).

Hellström [39] analyzed the steady-state conduction problem in the borehole cross-section in detail with the line source and multipole approximations. The line source assumption has resulted in the following solution.

$$\begin{cases} R'_{11} = \frac{1}{2\pi\lambda_g}\left(\ln\frac{r_b}{r_o} - \frac{\lambda_g - \lambda_s}{\lambda_g + \lambda_s}\ln\frac{4r_b^2 - D_u}{4r_b^2}\right) + \frac{1}{2\pi\lambda_p}\ln\frac{r_o}{r_i} + \frac{1}{2\pi h_i r_i} \\ R'_{12} = \frac{1}{2\pi\lambda_g}\left(\ln\frac{\sqrt{2}r_b}{D_u} - \frac{\lambda_g - \lambda_s}{2(\lambda_g + \lambda_s)}\ln\frac{16r_b^4 - D_u^4}{16r_b^4}\right) \\ R'_{13} = \frac{1}{2\pi\lambda_g}\left(\ln\frac{r_b}{D_u} - \frac{\lambda_g - \lambda_s}{\lambda_g + \lambda_s}\ln\frac{4r_b^2 + D_u}{4r_b^2}\right) \end{cases} \quad (8)$$

where, $\lambda_f$, $\lambda_g$, $\lambda_s$ and $\lambda_p$ are heat conductivity of fluid, grout, soil, and tube, respectively (W·m$^{-1}$·K$^{-1}$). $r_b$, $r_i$ and $r_o$ are borehole radius, internal and external radius of U-tube, respectively (m). $D_u$ is spacing between two legs of U-tube (m). $h_i$ is convection coefficient of the fluid inside tube (W·m$^{-1}$·K$^{-1}$).

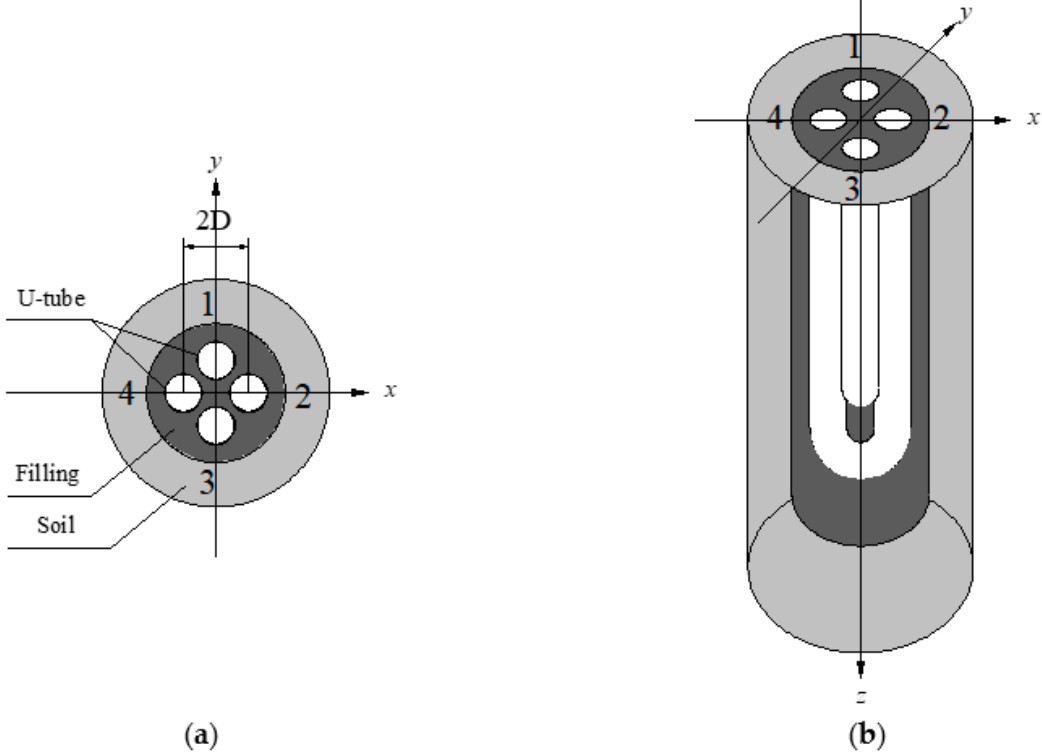

**Figure 3.** Schematic of a double U-tube BHE: (**a**) Two-dimensional (**b**) Three-dimensional.

The heat flux rate of each branch tube has a connection with the vertical temperature gradient of the fluid circulating in it. Assuming the vertical temperature of the tube in which the fluid flows upward to be positive and downward to be negative, the heat equilibrium of the fluid in each branch tube satisfies the following equations [38]:

$$
\begin{cases}
q_1 = -Mc_p \frac{\partial T_1}{\partial z} = K_1 \left[ T_{f1}(z) - T_b \right] + K_{12} \left[ T_{f1}(z) - T_{f2}(z) \right] + K_{13} \left[ T_{f1}(z) - T_{f3}(z) \right] \\
\qquad\qquad\quad + K_{12} \left[ T_{f1}(z) - T_{f4}(z) \right] \\
q_2 = -Mc_p \frac{\partial T_2}{\partial z} = K_{12} \left[ T_{f2}(z) - T_{f1}(z) \right] + K_1 \left[ T_{f2}(z) - T_b \right] + K_{12} \left[ T_{f2}(z) - T_{f3}(z) \right] \\
\qquad\qquad\quad + K_{13} \left[ T_{f2}(z) - T_{f4}(z) \right] \\
q_3 = Mc_p \frac{\partial T_3}{\partial z} = K_{13} \left[ T_{f3}(z) - T_{f1}(z) \right] + K_{12} \left[ T_{f3}(z) - T_{f2}(z) \right] + K_1 \left[ T_{f3}(z) - T_b \right] \\
\qquad\qquad\quad + K_{12} \left[ T_{f3}(z) - T_{f4}(z) \right] \\
q_4 = Mc_p \frac{\partial T_4}{\partial z} = K_{12} \left[ T_{f4}(z) - T_{f1}(z) \right] + K_{13} \left[ T_{f4}(z) - T_{f2}(z) \right] + K_{12} \left[ T_{f4}(z) - T_{f3}(z) \right] \\
\qquad\qquad\quad + K_1 \left[ T_{f4}(z) - T_b \right]
\end{cases}
\tag{9}
$$

where

$$
\begin{cases}
K_1 = 1/\left( R'_{11} + R'_{13} + 2R'_{12} \right) \\
K_{12} = R'_{12}/\left( R'^2_{11} + R'^2_{13} + 2R'_{11}R'_{13} - 4R'^2_{12} \right) \\
K_{13} = \left( R'^2_{13} + R'_{11}R'_{13} - 2R'^2_{12} \right)/\left[ \left( R'_{11} - R'_{13} \right) \times \left( R'^2_{11} + R'^2_{13} + 2R'_{11}R'_{13} - 4R'^2_{12} \right) \right]
\end{cases}
\tag{10}
$$

To simplify the expression in Equation (9), define $\theta_1(z) = T_{f1}(z) - T_b$, $\theta_2(z) = T_{f2}(z) - T_b$, $\theta_3(z) = T_{f3}(z) - T_b$, $\theta_4(z) = T_{f4}(z) - T_b$, and denote $S_1 = K_1/(Mc_p)$, $S_2 = (K_{12} + K_{13})/(Mc_p)$.

The two U-tube loops' temperature can be considered as equal for symmetry, that is, $T_{f1}(z) = T_{f2}(z)$, $T_{f3}(z) = T_{f4}(z)$. Equation (9) can be written in dimensionless form as

$$\begin{cases} -\frac{\partial \theta_1}{\partial z} = (S_1 + S_2)\theta_1 - S_2\theta_3 \\ \frac{\partial \theta_3}{\partial z} = (S_1 + S_2)\theta_3 - S_2\theta_1 \end{cases} \tag{11}$$

Solving the differential equations above, it yields,

$$\begin{cases} \theta_1(z) = C_1 \frac{S_1 + S_2 - \sqrt{(S_1+S_2)^2 - S_2^2}}{S_2} e^{(\sqrt{(S_1+S_2)^2 - S_2^2}) \cdot z} + C_2 \frac{S_1 + S_2 + \sqrt{(S_1+S_2)^2 - S_2^2}}{S_2} e^{-(\sqrt{(S_1+S_2)^2 - S_2^2}) \cdot z} \\ \theta_3(z) = C_1 \cdot e^{(\sqrt{(S_1+S_2)^2 - S_2^2}) \cdot z} + C_2 \cdot e^{-(\sqrt{(S_1+S_2)^2 - S_2^2}) \cdot z} \end{cases} \tag{12}$$

Subject to the boundary conditions:

$$\begin{cases} \theta_1|_{z=0} = \theta_1(0) = T_{in} - T_b \\ \theta_1|_{z=H} = \theta_3|_{z=H} = \theta_1(H) = \theta_3(H) \end{cases} \tag{13}$$

Therefore, the values of $C_1$ and $C_2$ can be obtained:

$$\begin{cases} C_1 = \frac{B_4 \cdot S_2 \cdot \theta_1(0)}{B_1 B_4 - B_2 B_3 B_5^2} \\ C_2 = \frac{B_3 \cdot B_5^2 \cdot S_2 \cdot \theta_1(0)}{B_2 B_3 B_5^2 - B_1 B_4} \end{cases} \tag{14}$$

$$\begin{cases} B_1 = S_1 + S_2 - \sqrt{(S_1+S_2)^2 - S_2^2} \\ B_2 = S_1 + S_2 + \sqrt{(S_1+S_2)^2 - S_2^2} \\ B_3 = S_1 - \sqrt{(S_1+S_2)^2 - S_2^2} \\ B_4 = S_1 + \sqrt{(S_1+S_2)^2 - S_2^2} \\ B_5 = e^{(\sqrt{(S_1+S_2)^2 - S_2^2}) \cdot H} \end{cases} \tag{15}$$

As a result, the outlet water temperature of the double U-tube BHE can be expressed as

$$\begin{aligned} T_{out} = \theta_3(0) - T_b &= C_1 + C_2 \\ &= \frac{B_4 \cdot S_2 \cdot \theta_1(0)}{B_1 B_4 - B_2 B_3 B_5^2} + \frac{B_3 \cdot B_5^2 \cdot S_2 \cdot \theta_1(0)}{B_2 B_3 B_5^2 - B_1 B_4} \\ &= \left( \frac{B_4 \cdot S_2}{B_1 B_4 - B_2 B_3 B_5^2} + \frac{B_3 \cdot B_5^2 \cdot S_2}{B_2 B_3 B_5^2 - B_1 B_4} \right) \cdot \theta_1(0) \end{aligned} \tag{16}$$

$\theta_1(0) = T_{in} - T_b$, it can be written further as

$$T_{out} = (T_{in} - T_b) \cdot \left( \frac{B_4 \cdot S_2}{B_1 B_4 - B_2 B_3 B_5^2} + \frac{B_3 \cdot B_5^2 \cdot S_2}{B_2 B_3 B_5^2 - B_1 B_4} \right) + T_b \tag{17}$$

The outlet water temperature of the buried tube can be obtained based on an accurate estimation of the borehole wall temperature $T_b$, which can be calculated using the model for the heat transfer process outside the borehole expressed by Equation (6) when the $r = \sqrt{r_b^2 + (z-h)^2}$.

Based on the above, the coupled analytical model with the heat transfer inside and outside the borehole can be solved through MATLAB programming. The flowchart of iterative solution procedure is as follows in Figure 4. On the premise of the inlet water temperature and given mass rate of the U-tube, a value for the outlet water temperature at a moment was assumed to be $T'_{out}$ and inputted to Equation (3). Then, an estimated value of borehole wall temperature $T_b$ was obtained. The value $T_b$ was inputted into Equation (17), which yields an updated U-tube outlet water temperature $T_{out}$.

This process was repeated until the difference between the estimated value and the updated value of the outlet water temperature meets the specified precision requirement. That is, by searching for the optimal solution, the actual $T_{out}$ at a certain moment was obtained when the value of $\left(T_{out} - T'_{out}\right)^2$ achieved the minimum value by using the MATLAB software, approaching zero.

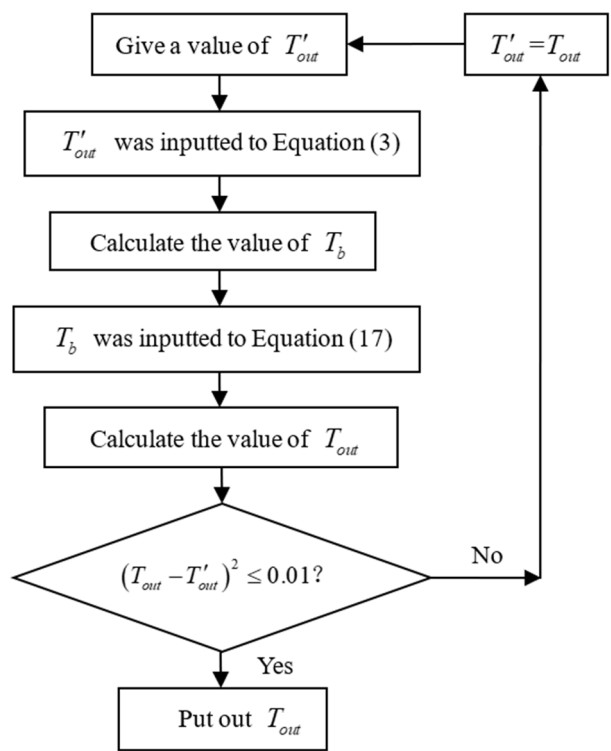

**Figure 4.** The flowchart of iterative solution procedure.

The heat transfer rate $q$ (W·m$^{-1}$) is one of the most important indices indicating the heat transfer performance of a BHE, and can be taken as an index to evaluate the heat transfer performance of a BHE and the operational efficiency of a GSHP system. It is defined as the following:

$$q = \frac{c_p \cdot M}{H}(T_{in} - T_{out}) \tag{18}$$

The heat transfer characteristics of a BHE can be assessed by the heat exchange effectiveness as well. The heat transfer effectiveness of a BHE, $\phi$, was defined as the ratio of the actual heat transfer rate $E$ to the ideal maximal value $E'$, expressed as the following,

$$\phi = \frac{\dot{E}}{\dot{E'}} = \frac{\dot{m}c(T_{in} - T_{out})}{\dot{m}c(T_{in} - T_0)} = \frac{T_{in} - T_{out}}{T_{in} - T_0} \tag{19}$$

where $T_0$ represents the initial soil temperature.

### 3.2. Model Validation

The data of the inlet and outlet water temperatures measured in an in situ geotechnical thermal response test were used to validate the calculation results of the coupled model established.

The geotechnical thermal response tests were conducted by constructing a double U-tube borehole. The basic parameters of the BHE are shown in Table 1 [40]. The LGGTP-1 type geotechnical thermal response in situ was used. The water flow rate and water temperature were measured by meters with deviations within ±0.5% and ±0.2 °C, respectively. The pump was turned on to circulate the water in

the loop. The stable inlet and outlet circulating water temperatures were measured. Their mean value was taken as the initial soil temperature at approximately 17 °C. Thereafter, the heating equipment was turned on to heat the circulating water at the rated power. The U-tube inlet and outlet water temperature were measured and recorded at a 10-min interval during the testing period.

**Table 1.** Basic parameters of the borehole with double U-tube tested.

| Parameter | Values | Parameters | Values |
|---|---|---|---|
| Borehole length/$H$ | 103 m | Density of water/$\rho$ | 1000 kg·m$^{-3}$ |
| Borehole radius/$r_b$ | 0.055 m | Specific heat of water/$c$ | 4200 J·kg$^{-1}$·K$^{-1}$ |
| U-tube inner radius/$r_i$ | 0.010 m | Water mass flow rate/$M$ | 0.51 kg·s$^{-1}$ |
| U-tube outer radius/$r_o$ | 0.0125 m | U-tube spacing/$D_u$ | 0.07 m |
| Soil thermal conductance/$\lambda_s$ | 3.08 W·m$^{-1}$·K$^{-1}$ | U-tube thermal conductance/$\lambda_p$ | 0.45 W·m$^{-1}$·K$^{-1}$ |
| Grout thermal conductance/$\lambda_g$ | 1.19 W·m$^{-1}$·K$^{-1}$ | Thermal conductance of tube fluid/$\lambda_f$ | 0.48 W·m$^{-1}$·K$^{-1}$ |

There is no obvious groundwater advection in the geological report of the geotechnical thermal response tests. It can only verify the analytical solution model for one working condition using the experimental data. The advection velocity was set to zero when the calculation of the coupled analytical model was carried out. The values of the double U-tube inlet water mass rate and dynamic inlet water temperature measured during the experiment were inputted into the established coupled model and the outlet water temperatures calculated correspondingly. The calculated values were compared with the data obtained in the experiment, as shown in Figure 5. The figure shows the deviations between the data monitored and calculated are less than 5% when the experimental data obtained in the initial eight hours were taken away. This finding indicates that a good agreement was achieved between experiment results and calculation results of the analytical heat transfer models.

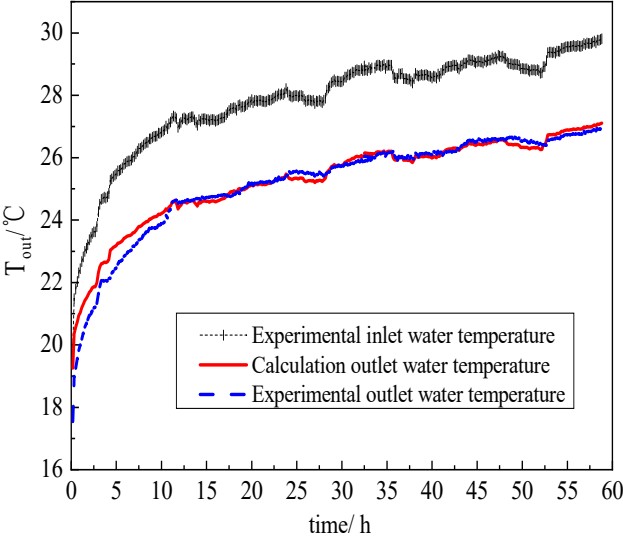

**Figure 5.** Comparison of the calculated and experimental values of the double U-tube BHE.

## 4. Results and Discussion

Groundwater advection velocity varies in a wide range because it is determined by the local hydraulic gradient and hydraulic conductivity in soils, which vary from place to place. In general, the Pelect number (hereafter referred to as *Pe*) can be taken as the criterion to determine whether the effect of the groundwater advection should be considered. Tan et al. [41] mentioned that when the *Pe* is in the range of 0.4–5, heat conduction and convection can be found in the soil; when the *Pe* is greater than 5, thermal convection dominates the heat transfer. However, the effects of groundwater advection on the underground loop design should be considered when the *Pe* is greater than 1 in

practical engineering. In the following, the effects of groundwater advection on the heat transfer performance of BHE in several types of soils were analyzed based on the calculation results of the proposed coupled model. The hydraulic gradient across most types of soils ranges between 0.0001 and 0.05 m·m$^{-1}$ [42]; the advection velocity in several kinds of soils based on the hydraulic gradient of 0.01 m·m$^{-1}$ was taken in the following. The corresponding physical parameters of the soils and the corresponding *Pe* are shown in Table 2 (Note: $Pe = \rho c(L/\lambda)$, $L$ = 4.5 m, typical borehole spacing). Groundwater advection velocities were found to differ even if the hydraulic gradients remain the same in different types of soils. The effects of groundwater advection in the silt and the sandstone can be ignored. Therefore, only the heat transfer performance of BHE in sand gravel, limestone, coarse sand, and fine sand were discussed in the following.

**Table 2.** Physical parameters of several typical soils with the same hydraulic gradient.

| Soil Media | Thermal Conductivity $\lambda$/W·m$^{-1}$·K$^{-1}$ | Volume Heat Capacity $\rho c$/J·m$^{-3}$·K$^{-1}$ | Permeability Coefficient $K$/m·s$^{-1}$ | Advection Velocity $u$/m·s$^{-1}$ | Porosity $\varepsilon$ | *Pe* |
|---|---|---|---|---|---|---|
| sand gravel | 0.98 | $1.4 \times 10^6$ | $3 \times 10^{-3}$ | $3 \times 10^{-5}$ | 0.31 | 578.57 |
| limestone | 3.56 | $1.34 \times 10^7$ | $1 \times 10^{-4}$ | $1 \times 10^{-6}$ | 0.275 | 5.3 |
| coarse sand | 1.02 | $1.4 \times 10^6$ | $7.3 \times 10^{-5}$ | $7.3 \times 10^{-7}$ | 0.385 | 13.53 |
| fine sand | 1.03 | $1.4 \times 10^6$ | $6.3 \times 10^{-6}$ | $6.3 \times 10^{-8}$ | 0.4 | 1.156 |
| silt | 2.07 | $2.85 \times 10^6$ | $1.4 \times 10^{-7}$ | $1.4 \times 10^{-9}$ | 0.475 | $1.28 \times 10^{-2}$ |
| sandstone | 4.5 | $3.56 \times 10^6$ | $4.2 \times 10^{-8}$ | $4.2 \times 10^{-10}$ | 0.18 | $1.76 \times 10^{-3}$ |

*Pe*—Pelect number.

In terms of the geometric and physical parameters of the double U-tube in BHE shown in Table 1, the outlet water temperatures were calculated based on the coupled model. The inlet water temperature was specified as 35 °C. The effects of the soil type, advection velocity, inlet water flow rate, and temperature on the outlet water temperature of the BHE were investigated in the following.

## 4.1. Effect of Soil Type

Table 2 shows that the groundwater advection velocities in different types of soils differ considerably while the hydraulic gradients are the same. The calculated values of the outlet water temperature of the double U-tube BHEs in sandy gravel and other four kinds of soils with a hydraulic gradient of 0.01 m·m$^{-1}$ are illustrated in Figure 6. The outlet water temperatures of the BHE in every kind of soil tend to rise and then stabilize. However, the change rates of the outlet water temperatures and the stable values achieved vary greatly in different kinds of soils.

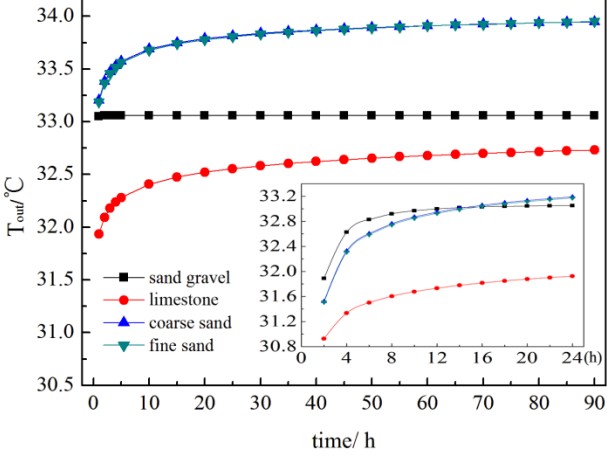

**Figure 6.** Dynamic outlet water temperatures in different types of soils.

The heat transfer efficiency of different soil is commonly decided by the thermal diffusion coefficient and the groundwater advection velocity. Figure 6 shows that the groundwater advection has not played a major role in the initial heat dissipation period and that the outlet water temperature from the BHE in soil with a small value of thermal conductivity rise very rapidly. As time elapses, the outlet water temperature of the BHE in sand gravel levels off very quickly and tends to reach a small value compared with those in coarse sand and fine sand. This finding can be attributed to the small thermal conductivity of sand gravel while the corresponding groundwater advection velocity is large, which can accelerate heat migration further. For instance, the outlet water temperature of the BHE in the limestone is the lowest, because its thermal conductivity is about four times the value of the sand gravel, and the groundwater advection velocity in the limestone is not small, thereby enhancing the convective heat transfer ($Pe \approx 5$). As a result, the temperature of the limestone surrounding the BHE did not rise dramatically. It takes about 12 h, 36 h, 52 h, or 53 h for the outlet water temperatures of the BHE in sand gravel, limestone, coarse sand, or fine sand to stabilize, respectively. In addition, an increase in groundwater advection velocity in the soils causes the outlet water temperature to become stable at a shorter time.

Figure 7 illustrates the dynamic change in heat transfer effectiveness of the double U-tube BHE in several typical soils as shown in Table 2. The heat transfer effectiveness decreases gradually until it tends to different stable values in different types of soils. In Figure 7a, for instance, the heat transfer effectiveness is 0.108, 0.17, 0.099, and 0.1 after a 24-h continuous heat rejection of the BHE in the four typical types of soils, respectively. They become 0.107, 0.134, 0.064, and 0.06, respectively after 30 days of continuous heat rejection. The heat transfer effectiveness of the BHE in limestone is the greatest and has the largest value of thermal conductivity. The heat transfer effectiveness decreased faster in the soil with small groundwater advection velocity and the stable value is affected by groundwater advection but also depended on thermal physical properties of the soil. That is, the soil with a large value of groundwater advection velocity and a large thermal conductivity can alleviate the heat accumulation in the soil and maintain the heat transfer effectiveness of the double U-tube heat exchanger at high value for a long time.

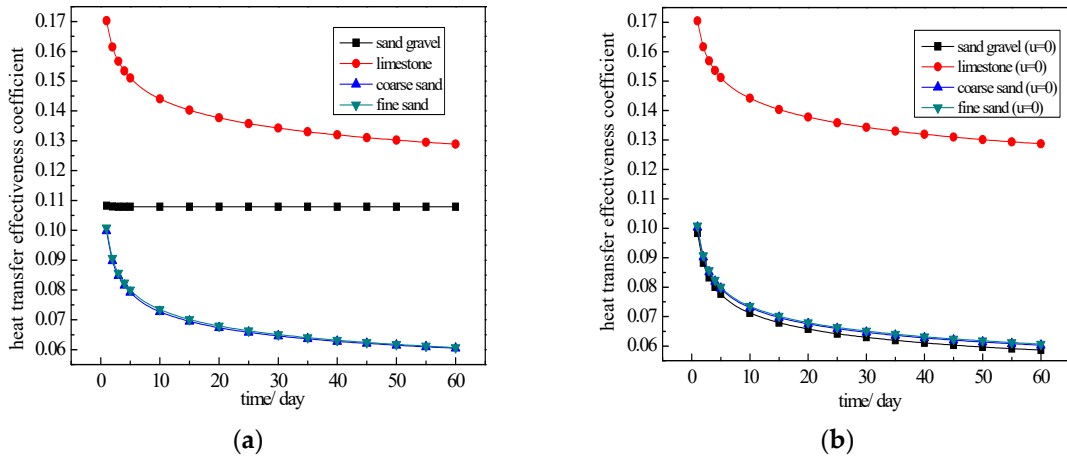

**Figure 7.** Transient variations of heat transfer effectiveness of the double-U tube BHE in different typical soils. (**a**) Effect of groundwater advection (**b**) Effect of groundwater advection ignored.

Figure 7a,b shows that the heat transfer effectiveness coefficient decreased rapidly as time elapsed in the sand gravel with groundwater advection in comparison with soils without groundwater advection because the smaller the heat conductivity coefficient of the sand gravel, the greater the advection velocity. As a result, heat convection dominated the heat transfer process. Because the limestone has large advection velocity and its heat conductivity is not small, the heat convection effect generated by advection did not dominate the heat transfer process. Moreover, the differences in heat

transfer effectiveness in the soils with and without groundwater advection were not significant because the advection velocities in coarse sand and fine sand are small and, thus, its effect can be ignored.

Then, taking the sand gravel as an example, the groundwater advection velocity is the largest among the soils listed in Table 2 and the outlet water temperatures of the double U-tube were compared with those from the double U-tube buried in sandy gravel without groundwater advection, as shown in Figure 8. It can be seen that the outlet water temperature rose comparatively faster in the sand gravel with groundwater advection and the temperature difference between the in-tube water and the soil decreased rapidly at the initial stage because heat convection strengthened the heat transfer and alleviated the heat accumulation around the BHE. As a result, the outlet water temperatures were comparatively high in the sand gravel with groundwater advection. As time elapsed, the temperature difference between the in-tube water and the soil increased and the heat transfer was enhanced, causing the outlet temperature to approach a low stable value very fast. After a 120 h continuous heat rejection, the outlet water temperature of the double U-tube in the sand gravel with groundwater advection was 0.54 °C lower than the value in the case of no groundwater advection. Simultaneously, the heat transfer rate of the double U-tube $q$ increased by 22.63 W·m$^{-1}$ to the value of 80.8 W·m$^{-1}$, which implies that a deviation in the heat transfer rate of up to 38.9% would be achieved if the existence of groundwater advection were ignored.

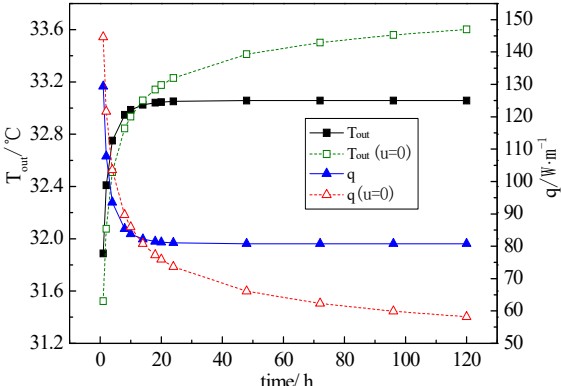

**Figure 8.** Transient variations of outlet water temperature and heat transfer rate in sand gravel.

### 4.2. Effect of Groundwater Advection Velocity

The underground hydraulic characteristics vary dramatically in different types of soils, and, thus, analyzing the heat transfer performance of a BHE in different types of soils with the different hydraulic gradients in the normal range (0.0001–0.05 m·m$^{-1}$) is important. Figure 9 reveals how the stable values of the heat transfer rates vary with the groundwater advection velocity in various types of soils surrounding the double U-tube. A large advection velocity brings a large value of the heat transfer rate, which means an increase in groundwater advection velocity will cause the heat transfer capability to increase in certain soils. The increasing trends of the heat transfer rates are consistent in the sand gravel, coarse sand, and fine sand with groundwater advection because the comprehensive thermal conductivity and the volumetric specific heat of these three types of soils discussed are roughly the same, but their porosity and permeability are not the same in practice, leading to different values of groundwater advection velocity.

Figure 9 shows that the heat transfer rate of the double U-tube in limestone is greater than that in silt when the groundwater advection velocity is less than $5 \times 10^{-6}$ m·s$^{-1}$. It becomes greater in the limestone than that in the silt when the groundwater advection velocities exceed $5 \times 10^{-6}$ m·s$^{-1}$ because the heat conductivity of the limestone is greater than that of the silt, and heat conduction dominates the heat transfer between the borehole and the surrounding soil when the groundwater advection velocity is small. The increase in the advection velocity may enhance the convection heat transfer and the silt has a greater value of thermal diffusivity than that of the limestone, and, thus, the heat can spread out

faster in the silt than it does in the limestone. Sandstone has the largest value of thermal conductivity among all types of soils discussed. Heat conduction dominates the heat transfer in sandstone even if the groundwater advection is obvious. As a result, the heat transfer rate in the sandstone is the largest among all soils mentioned above.

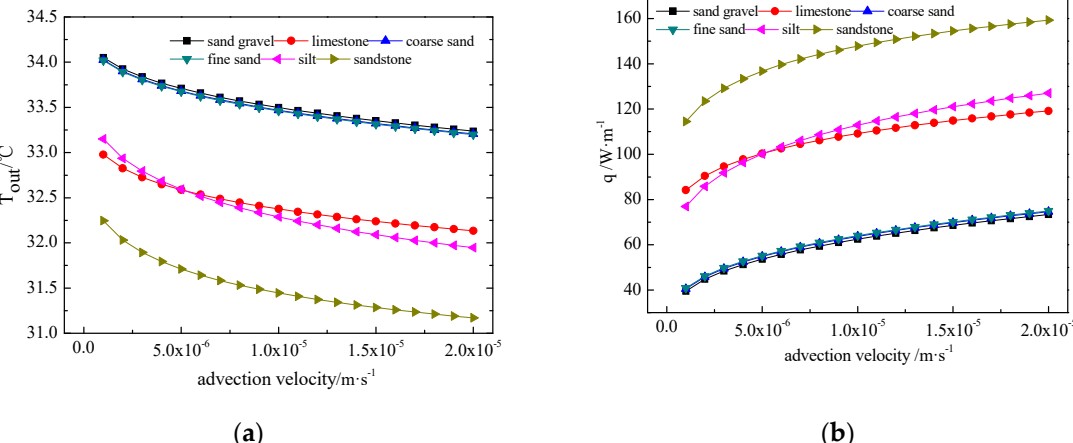

**Figure 9.** Variations of outlet water temperature and heat transfer rate with groundwater advection velocities in different types of soils. (**a**) Outlet water temperature (**b**) Transfer rate.

### 4.3. Effect of Water Mass Flow Rate

When the other conditions are the same, the different water mass flow rates will cause the dynamic outlet water temperature of the double U-tube to vary in the same performance period. Figure 10 illustrates how the transient outlet water temperature and heat transfer rate in the sand gravel vary in the cases of different water mass flow rates when the inlet water temperature is fixed at 35 °C. The outlet water temperature reaches 30.48 °C, 31.82 °C, 32.56 °C, 33.01 °C, and 33.33 °C after 24-h continuous heat transfers, in the cases of the mass flow rate being maintained at 0.2 kg·s$^{-1}$, 0.3 kg·s$^{-1}$, 0.4 kg·s$^{-1}$, 0.5 kg·s$^{-1}$, and 0.6 kg·s$^{-1}$, respectively. This finding implies that a large inlet mass flow rate causes an increase in the outlet water temperature of the U-tube. However, the amplitude of the increases tends to be smaller as the increase in the mass flow rate continues because the increase in the water mass flow rate shortens the time it takes for the water to flow through the U-tube loop. Meanwhile, the in-tube heat transfer will increase as the water mass flow rate increases. As a result, the heat transfer rate will be improved. The amplitude of the increment decreases as the water mass flow rate increases. That is to say, the turbulence becomes more intense in the U-tube as the inlet mass flow rate increases, and the convective heat transfer coefficient between the water in the tube and the borehole wall will also increase and that leads to an increase in heat exchange capacity. Therefore, in the design stage of BHE, the water mass flow rate must be chosen properly, otherwise, the heat transfer rate may become very low or the pump power consumption will become large. Thus, the mass flow rate should be as large as possible to generate or enhance turbulence in the tube to facilitate the heat transfer. It should not be too large so as control the power consumption of circulating pumps within a reasonable range. In practical engineering applications, the size of the BHE given in this paper is a common size; the recommended inlet water mass flow rate through the double U-tube BHE should be more than 0.5 kg·s$^{-1}$, but not exceed a certain upper limit otherwise the gain outweighs the gain.

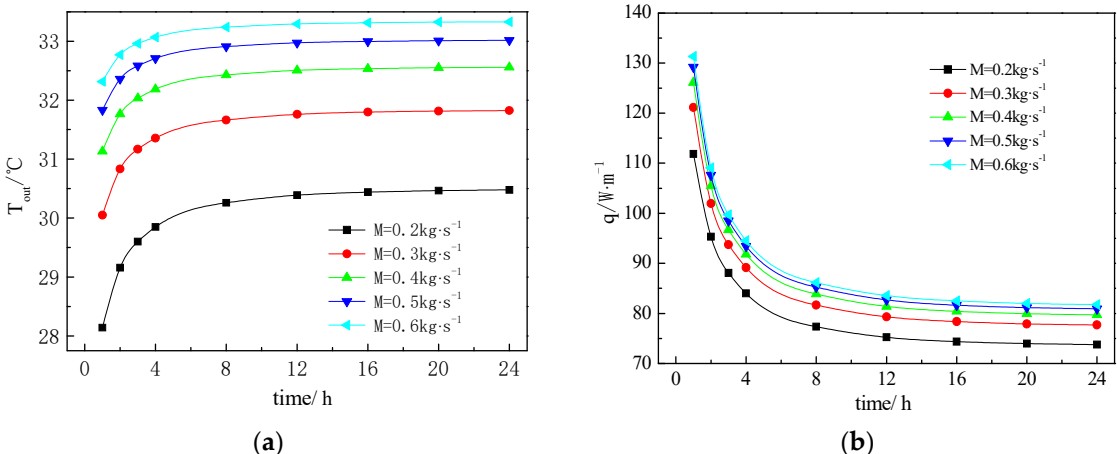

**Figure 10.** Dynamic variations of outlet water temperature and transfer rate in the cases of different in-tube water mass flow rates. (**a**) Outlet water temperature (**b**) Transfer rate.

### 4.4. Effects of Inlet Water Temperature

Different inlet water temperatures of the U-tube will result in different dynamic outlet water temperatures. Figure 11a,b shows the dynamic outlet water temperatures and heat transfer rates under the conditions of different inlet water temperatures with the same water mass flow rate of 0.5 kg·s$^{-1}$ to the double U-tube BHE in the sand gravel. Figure 11a shows that the outlet water temperatures reached 27.70 °C, 30.38 °C, 33.05 °C, or 35.73 °C, respectively, after a 24-h continuous heat rejection of the BHE in the sand gravel when the inlet BHE temperatures were 29 °C, 32 °C, 35 °C, or 38 °C. Figure 11b shows that the corresponding heat transfer rates were 54.03 W·m$^{-1}$, 67.53 W·m$^{-1}$, 80.04 W·m$^{-1}$, or 94.54 W·m$^{-1}$, respectively. This result indicates that the rise in the inlet water temperature causes the temperature difference between the inlet and outlet water to increase while the water mass rate remains constant. As a result, the heat transfer rate increases. That is, high inlet temperature causes a large value of heat transfer rate. Moreover, the higher the inlet water temperature, the larger the difference between the water temperature inside U-tube and the initial soil temperature. Thus, the heat transfer effectiveness of the BHE with double U-tube at different inlet temperatures is maintained at about 0.108 in the cases we studied. Thus, the effect of the inlet water temperature of U-tube on heat transfer effectiveness is insignificant.

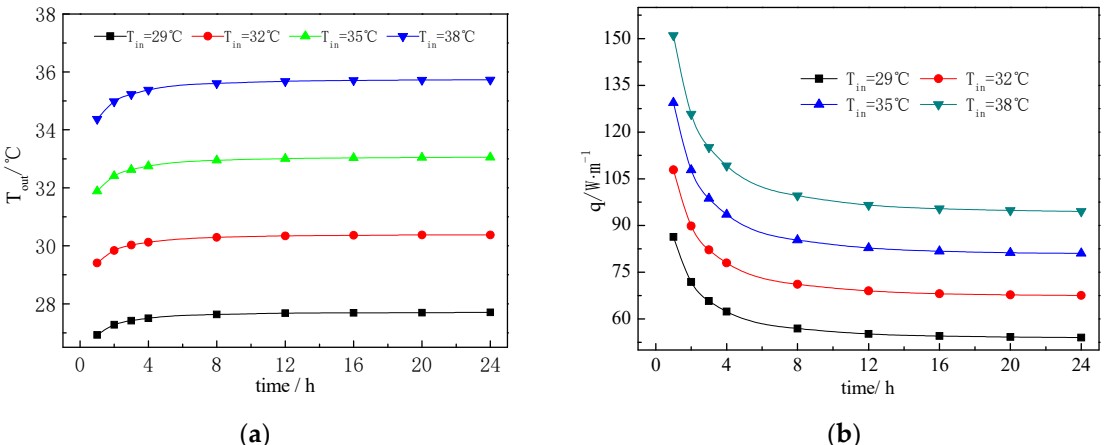

**Figure 11.** Dynamic outlet water temperatures and transfer rates for different inlet water temperatures. (**a**) Outlet water temperature (**b**) Transfer rate.

## 5. Conclusions

The current GHE analytical model that considers the effects of groundwater advection cannot solve the coupled heat transfer problem from the in-tube fluid to the soil with groundwater advection comprehensively. Hence, the proposed coupled model is an improvement of the previous analytical solution for the double U-tube because it considers heat transfer inside a borehole. The coupled model can be applied for a quick and easy estimation of the outlet temperature that considers the effects of groundwater advection in some mediums. The following conclusions can be drawn:

(1) The heat convection between the buried tube and soil is reinforced with the increase in groundwater advection velocity in the soil. As a result, the outlet water temperature of the buried U-tube drops and the heat transfer rate increases. Thus, clarifying the in situ groundwater flow conditions in the soil is important before designing a BHE.

(2) The heat transfer process of the buried U-tube in several typical soils with groundwater advection was discussed. The thermal conductivity of soil plays a leading role in heat transfer at the initial early stage. The effect of groundwater advection appears gradually as time goes on. A large value of advection velocity is brought about in the short time it takes for the outlet water temperature of the buried U-tube to achieve stability. Hence, the length of the buried tube designed can be reduced and the initial investment reduced if groundwater advection velocity can be larger than a certain value.

(3) With the increase of inlet mass flow rate, the turbulence intensity in the tube increases and the total heat transfer of double U-tube augment but the enlarge of the flow resistance leads to the increase of pump power consumption. Therefore, the influence of heat exchange and power consumption should be considered in the choice of inlet mass flow rate. The effects of the inlet temperature to the buried U-tube on the heat transfer effectiveness of BHE can be neglected.

**Author Contributions:** Conceptualization and methodology, L.Z.; software and simulation, T.Y. and Z.S.; original draft preparation, L.Z. and Z.S.; review and revision Z.L. and T.Y. All authors have read and agreed to the published version of the manuscript.

**Funding:** This work was supported by the National Key Research and Development Program of China during the Thirteenth Five-year Plan (grant No.2018YFD1100705-04).

**Conflicts of Interest:** All authors declare there is no conflict of interest. The MATLAB code of the heat transfer model proposed in this paper can be obtained freely by the URL: https://github.com/yth0918/ground-heat-transfer-with-groundwater-advection/blame/1c6f795357ff1c4e98572a8c5a1b51a0e15fbad2/README.md.

## Nomenclature

| | |
|---|---|
| $c_p$ | Specific heat capacity, $J \cdot kg^{-1} \cdot K^{-1}$ |
| $D_u$ | Spacing between two legs of U-tube, m |
| $E, E'$ | The ratio of the actual and the ideal heat transfer to the maximal heat transfer |
| $H$ | Borehole depth, m |
| $h_i$ | Convection coefficient of the fluid inside tube, $W \cdot m^{-1} \cdot K^{-1}$ |
| $K$ | Permeability coefficient, $m \cdot s^{-1}$ |
| $L$ | Characteristic length, m |
| $M$ | Mass flow rate of water, $kg \cdot s^{-1}$ |
| $n$ | Correction factor, heating $n = 0.4$, cooling $n = 0.3$ |
| $q$ | Heat transfer rate, $W \cdot m^{-1}$ |
| $R'_{11}$ | Thermal resistance between the fluid in each U-tube leg and the borehole wall, $m \cdot K \cdot W^{-1}$ |
| $R'_{12}$ | Thermal resistance between two adjacent U-tube legs, $m \cdot K \cdot W^{-1}$ |
| $R'_{13}$ | Thermal resistance between two symmetric U-tube legs, $m \cdot K \cdot W^{-1}$ |
| $r_b, r_i, r_o$ | Borehole radius, internal and external radius of U-tube, m |
| $T$ | Temperature, °C |
| $U$ | Equivalent advection velocity, $m \cdot s^{-1}$ |
| $u$ | Advection velocity, $m \cdot s^{-1}$ |
| $a$ | Thermal diffusivity, $m^2 \cdot s^{-1}$ |

## Nomenclature

| | |
|---|---|
| $\varepsilon$ | Porosity |
| $\lambda_f, \lambda_g, \lambda_s, \lambda_p$ | Heat conductivity of fluid, grout, soil and tube, W·m$^{-1}$·K$^{-1}$ |
| $\rho$ | Density, kg·m$^{-3}$ |
| $\tau$ | Time, s |
| $\varphi$ | heat transfer effectiveness coefficient |

**Subscripts**

| | |
|---|---|
| 0 | initial |
| $s$ | soil |
| $w$ | water |
| $in$ | inlet |
| $out$ | outlet |
| $f_1, f_2$ | the downward and upward U-tube legs |

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
