# Peer review of "Study on the Coupled Heat Transfer Model Based on Groundwater Advection and Axial Heat Conduction for the Double U-Tube Vertical Borehole Heat Exchanger"

_sustainability, doi:10.3390/su12187345_

Round 1
Reviewer 1 Report
The manuscript is well-organized and clear. Details of analytical solutions are recommended to be added in an appendix. The topic is relevant but the contribution of the paper is limited. Although the analytical solution seems soundness, the model is only compared with one experimental case in a very limited way in Fig 2. Results and discussion are basic sensitivity analyses of the analytical solution. The results and conclusions are reasonable, with some suggestion/comments to be addressed:
-The reviewer suggests expanding the introduction to explain more the applicability of a GSHP system and then achieve a larger public.
-The word system is written twice in line 39.
- GHE needs to be defined in line 41.
- The authors should use either GSHPS or GSHP system to be consistent.
- In line 48, it should be "include a line and cylindrical heat source".
- Need to add references to the commercial tools in line 58.
- Line 109, "BHE designers."
- line 123, "The heat flux... depends"
- It would be nice to have a figure showing the boundary conditions explained in the first paragraph of section 3.1.
- The sentence in lines 151-153 is a little confusing because the numbers seem to refer to Fig. 1 but the wording is general.
- Figure 2 legend is blurry but it seems that the calculated inlet temperature is aligned with the experimental outlet temperature and, therefore, there is something off.
- In line 209, the sentence "No obvious groundwater advection (was) found in the geological report. Thus, the advection velocity was set to zero" seems to be in contradiction with the introduction (line 93) "Thus, considering the effects of groundwater advection is important in the improvement of the BHE design..."
- In Section 4.3, the increase of mass flux shouldn't increase the outlet temperature unless your heat source is increasing. The authors may need to revise their boundary conditions or make the model limitation clearer.
- A comparison with only one experimental condition does not validate a model for all the range of conditions discussed in the results section. It only suggests a good agreement. The authors are highly recommended to revise their wording.
Reviewer 2 Report
This manuscript proposed an analytical model that considers the effect of groundwater advection coupled heat transfer problem. The manuscript is well written and organized. The analysis is performed in both depth and width. The conclusions are pertinent and clear. I would recommend acceptance for this manuscript. I have one suggestion for this manuscript. It is better to have a figure illustrating the whole system to give a general background of this study.
Reviewer 3 Report
The focus of the manuscript is a developed mathematical model on heat transfer in BHE. The manuscript reads very technical for the journal "Sustainability", though this is a decision to be made by the editor. The manuscript is, in principle, well written and structured. Honestly, I am missing a degree of novelty as the importance of groundwater flow for BHE has been examined extensively before. My assessment would change, if the authors would make their computer model freely available (e.g. upload to github or any other source code repository) in combination with an explanation of the source code, exemplary explanation of the handling of the model and a well-written source code itself! Right now, it is just another model and a parameters study of this model with little novelty and unreproducible results for anyone else.
The mathematical model in principle looks profound and correct BUT there are major and fundamental issues that need to be addressed:
- The model description is terrible. There is not ONE systematic sketch. The geometry of the model is not explained properly. Symbols used are not explained or even used twice. Units of quantities are not given. The used principle of moving finite line heat source is not properly explained. The mathematics is complex and derivations are not presented nor there are any references for the theoretical part. It therefore remains unclear which parts of the model are from the authors and which are from previous works. This section does not need to be improved, it needs to be re-written with a clear structure, a clear geometrical representation, a description of the case and geometry considered and then a mathematical model on top of that!
- The assumptions of the theoretical and numerical model are neither mentioned nor discussed.
- Parameterization of the model remains unclear, as the whole model is unclear. Consider including an extra section regarding parameterization.
- The validation case is not properly discussed (might correspond to the lack of description of the model itself).
- The "Results" section is a pure parameter variation. The scientific outcome is low.
- The drawn conclusions are partly not supported by the results.
- Statements in the abstract do not correspond to the discussion/conclusion section of the manuscript.
For minor changes, please see the attached pdf.
I recommend the authors, to re-submit the manuscript with a different focus: The mathematical model itself seems properly done and the paper itself is not badly written. So take the time to clean up the model description extending your theoretical derivations, describe how you numerically designed you model using Matlab, clean up your source code following respective standards (line indentation, comments in the source code, naming conventions...), upload your code to an open repository such as gitlab/github or any other, include a section on how to use this code in your manuscript (what function does what, relevant parameters and so on). Then streamline your results/discussion and especially your conclusions to match the statements made in conclusions and abstract. And this will be a proper manuscript to be accepted.

Round 2
Reviewer 1 Report
There were significant improvements after the revision. However, it is still confusing why the authors use experimental data where the groundwater advection is negligible if they want to show the importance of the advective term.
The effect clear appears on their sensitivity analysis.
Some editorial comments:
L24-25: The efficiency of the heat transfer coefficient of the double U-tube BHE was determined...
L38: energy-saving
L47: The depth...
L112: of the various heat transfer mechanisms
L178: The simplifications are explained as follow:
L263-265: Very confusing sentence...
L393-396: This sentence is too long. Break into two to be clearer.
Author Response
Please see the attachmen.

Reviewer 3 Report
Thank you very much for this revision! The authors clearly did an amazing job to improve the manuscript in so many fashions! Now it is really possible to understand your work.
There has been one misunderstanding though, which I would like to clarify and I would like the authors to think of: I did not mean to make the Matlab code available to ME(!) as a reviewer (and your code looks fine) but to make it freely available to everyone by uploading it to an open repository such as gitlab, github or any other repository (it might be that MDPI even provides one). In this case, you will see that your citations will go high quickly because many other researchers will use your model! This would really benefit your work (and career)!
Considering this, I am glad to recommend the publication of your work.
There are minor spelling errors. Please check (e.g.)
- line 112: it is complicated TO MODEL(?)
- line 176: is establishED
- line 178: TheY are
Author Response
Please see the attachmen.